Variations in the reproductive strategies of three populations of Phrynocephalus helioscopus in China

Liang Tao 1
Zhou Lu 1 2
He Wenfeng 1
Xiao Lirong 1
Shi Lei leis@xjau.edu.cn shileixj@126.com 1
1 College of Animal Science, Xinjiang Agricultural University , Urumqi , Xinjiang , China
2 Institute of Hydrobiology, Chinese Academy of Sciences , Wuhan , Hubei , China
Roper James
Electronic publication date: 2018 Oct 24
Publication date: 2018
Volume: 6
Electronic Location ID: e5705
Received 2018 Apr 13; Accepted 2018 Sep 6
Copyright: ©2018 Liang et al.
Copyright year: 2018
Copyright holder: Liang et al.
License: This is an open access article distributed under the terms of the Creative Commons Attribution License, which permits unrestricted use, distribution, reproduction and adaptation in any medium and for any purpose provided that it is properly attributed. For attribution, the original author(s), title, publication source (PeerJ) and either DOI or URL of the article must be cited.
License URL: https://creativecommons.org/licenses/by/4.0/

Keywords: Phrynocephalus helioscopus, Optimal egg size, Morphological constraint hypothesis, Egg size-number trade-off, Life history, Reproduction

Funding: National Natural Science Foundation of China 31660613 The research is funded by the National Natural Science Foundation of China (31660613). The funders had no role in study design, data collection and analysis, decision to publish, or preparation of the manuscript.

==============================
Background

Egg size and clutch size are key life history traits. During the breeding period, it is possible for females to increase their reproductive output either by increasing the number of eggs if the optimal egg size (OES) is maintained, or by increasing the allocation of energy to each egg. However, the strategies adopted are often influenced by animals’ morphology and environment.

Methods

Here, we examined variation in female morphological and reproductive traits, tested for trade-offs between egg size and clutch size, and evaluated the relationship between egg size and female morphology in three populations of Phrynocephalus helioscopus.

Results

Female body size, egg size, and clutch size were larger in the Yi Ning (YN) and Fu Yun (FY) populations than in the Bei Tun (BT) population (the FY and YN populations laid more, and rounder eggs). Egg size was independent of female body size in two populations (BT and FY), even though both populations had an egg-size/clutch size trade-off. In the YN population, egg size and clutch size were independent, but egg size was correlated with female body size, consistent with the hypothesis of morphological constraint.

Conclusions

Our study found geographical variation in body size and reproductive strategies of P. helioscopus. Egg size was correlated with morphology in the larger-bodied females of the YN population, but not in the smaller-bodied females of the BT population, illustrating that constraints on female body size and egg size are not consistent between populations.

Background

Reproductive traits are often variable in animals as a result of differences in the quality of resources and food availability in different habitats (Roff, 2002; Cruz-Elizalde & Ramırez-Bautista, 2016). Egg size and clutch size are key life history traits, and have received more attention than other reproductive traits (Amat, 2008; Qu et al., 2011; Lovich et al., 2012). When there is less food available, females may face the problem of limited available reproductive resources to invest in eggs. This results in a trade-off between (1) the energy allocated to each egg (egg size), and (2) the total number of eggs (clutch size). An increase in resources allocated to each egg will result in a decrease in clutch size (Roff, 1992; Kaplan & Phillips, 2006). This negative relationship between egg size and clutch size provides evidence for reproductive trade-offs (Rowe, 1992). Variation in female reproductive output is widespread, both interspecifically and intraspecifically. Especially for geographically widespread species, local genetic variation, short-term phenotypic plasticity, and the complex interactions between these two, contribute to variation in reproductive output (Brown & Shine, 2007).

Optimal egg size (OES) theory predicts that natural selection optimizes egg size within populations, such that when resources are not limiting for reproduction, clutch size or number of clutches may increase, while egg size remains constant (Smith & Fretwell, 1974; Brockelman, 1975). Natural selection predicts that females should optimize resources allocated to each egg, and clutch size should only increase after ensuring the production of high quality offspring (Lovich et al., 2012). In some reptiles, CS is positively correlated with female body size, while egg size remains constant, consistent with the OES theory (Congdon & Gibbons, 1987). However, the relationship between egg size and clutch size is determined by many factors, and the trade-offs between egg size and clutch size are not always evident in natural populations (Berven, 1982; Liao & Lu, 2011; Wang et al.,  2011).

Egg size is often correlated with female body size in reptiles (morphological constraint hypothesis), and both egg size and clutch size increase with an increase in female body size, contrary to OES theory (Dunham & Miles, 1985; Clark, Ewert & Nelson, 2001; Ryan & Lindeman, 2007; Mohamed et al., 2012). When resources are limited, reproductive output is directly correlated with the trade-offs between egg size and clutch size, and ultimately, with offspring survival (Congdon & Tinkle, 1982; Brown & Shine, 2009). The size of each egg normally determines the success of incubation and offspring survival (Angilletta et al., 2004; Räsänen, Laurila & Merilä, 2005). Females may increase energy allocation to eggs as a way to improve offspring quality.

P. helioscopus are small (mean snout-vent length (SVL): 47.5 mm) lizards that are widely distributed in Eurasia. Previous research on this species has focused on egg incubation (Wang et al., 2013), and female reproductive output (Liang et al., 2015). However, variation in female reproductive traits, and egg size-clutch size trade-offs, and the effects of female size on egg size have not been studied in geographically different populations in order to understand variation in investment in offspring production. In this study, we compared female morphological traits and the relationships among egg length (EL), egg width (EW), egg mass (EM), egg shape (ES), and clutch size (CS) in three populations of P. helioscopus. Specifically, we:

1. Tested whether reproductive female size differed between the three populations, and

2. Examined how that variation was associated with reproductive traits, especially fecundity, egg and clutch size, egg shape, and the trade-offs between egg size and clutch size.

Materials and Methods

Ethical approval

Specimens were collected following Guidelines for Use of Live Amphibians and Reptiles in Field Research (the Herpetological Animal Care and Use Committee (HACC) of the American Society of Ichthyologists and Herpetologists, 2004). This study was conducted in compliance with current laws on animal welfare and research in China and the regulations set by the Xinjiang Agricultural University. After the research was completed, the lizards were released where they were captured.

Study site

The populations studied here are located in three ecologically distinct locations in the Xinjiang Uyghur Autonomous Region, China, and included locations near Bei Tun city (BT: 87°15″E, 47°26′N), Fu Yun city (FY: 89°05′E, 46°36′N), and Yi Ning city (YN: 80°47′E, 43°40′N). The BT and the YN populations were located approximately 660 km apart, and the habitats were different at each location. The BT population was located in a typical gravel desert with little vegetation, while the YN population was located in a loam desert with abundant vegetation. The climate experienced by the YN population was hotter and wetter than that experienced by the BT population. The FY and BT populations were located approximately 160 km apart, and the FY and YN populations were located approximately 700 km apart. The vegetation and rainfall at the FY and YN locations were similar, while the temperature regimes were similar at the FY and BT locations (Figs. 1 and  2).

Figure 1 Map showing the three locations in the Xinjiang Uyghur Autonomous Region of western China where P. helioscopus were captured for this study.

The nearest cities (BT, Bei Tun, FY, Fu Yun; YN, Yi Ning) are identified by the red dots, and collection locations are indicated by black dots with arrows. Photos indicate habitat types in each sampling location (Photo credit: Tao Liang).

Figure 2 Means for monthly mean air temperature (A) and monthly mean rainfall (B) from 1990 to 2013 at three sampling locations (BT: Bei Tun, FY: Fu Yun, YN: Yi Ning) where P. helioscopus were collected.

Note that months are numbered from 1 (January) to 12 (December).

Animal and egg collection

In May 2014 and May 2017, we collected P. helioscopus individuals by hand from locations near BT (in 2014 Liang et al., 2015), FY (in 2017), and YN (in 2017). We collected lizards from 12:00 to 18:00, when they were most active (T Liang, pers. obs., 2015, 2017; S Lei, pers. obs., 2015), and we captured the lizards at random. Lizards were transported to Xinjiang Agricultural University, where female lizards were palpated to determine their reproductive state (Li et al., 2006). Fifty-three gravid females (BT: 13, FY: 24, YN: 16) were housed individually in plastic cages containing vermiculite in a room with ambient temperatures that varied from 20–28 °C with a 12-hour light/12-hour dark cycle. A 250 W light bulb was suspended at one end of each cage, 20 cm above the cage floor, and the lizards could freely move to warmer and cooler places within the cage. Mealworms (larvae of Tenebrio molitor) and water enriched with vitamins and minerals were provided ad libitum. Females dig before they lay eggs, and we observed females every 2 hours, which allowed us to collect eggs quickly, and prevented eggs from absorbing water in the moist vermiculite. All eggs used in this study were collected no more than 20 min after they were laid.

Morphology and reproductive traits

We measured female snout-vent length (SVL), tail base width (TBW), egg length (EL), and egg width (EW), using digital calipers. All measurements were accurate to within 0.1  mm. We also recorded the clutch size (CS). We weighed eggs (egg mass, EM) and clutches (clutch mass, CM) on an electronic balance to the nearest 0.01 g. The ratio of egg length to egg width (EL/EW) indicates the general shape of the eggs (egg shape, ES), where 1 is a round egg, and larger values indicate increasingly elongate eggs (Ji & Wang, 2005; Kratochvíl & Frynta, 2006).

Statistical analyses

We used analysis of variance (ANOVA) to examine differences in SVL, EM, and ES, and we used analysis of covariance (ANCOVA) to examine differences in TBW, EL, EW, RCM, and CS between the three populations with post hoc Tukey’s tests (multiple comparisons). We used Levene’s tests to test for heterogeneity of variances. The residuals were tested by Kolmogorov–Smirnov tests for detecting normality. We log-transformed the variables to minimize the heterogeneity, where necessary (King, 2000). To test egg size-clutch size trade-offs, and analyze potential morphological constraints on optimal egg size, the relationships between EM and EL and CS, EM and EL, EL and CS and SVL, and EW and TBW were examined using reduced major axis (RMA) regressions rather than ordinary least squares (OLS) regressions, because RMA accounts for an error in the independent variable (Dunham & Miles, 1985). Historical climatic data (1990–2013) of the three study areas were taken from the Chinese National Climatic Data Center (http://data.cma.cn). Descriptive statistics are presented as mean adjusted (calculated using the effect function in the “effects” package, (Fox & Hong, 2009) ± standard error (SE), except for SVL, EM, and ES, which are presented as the mean ±SE. Differences were considered significant when P < 0.05.

All analyses were conducted using R v. 3.4.1 (R Core Team, 2017), using the packages “lmodel2” (Legendre, 2011), “ggplot” (Wickham, 2015), and “gplots” (Warnes et al., 2011).

Results

Female morphological variation

SVL varied among populations, and was longest in the YN and FN populations, in which females had similar SVLs (YN: 51.23 mm; FY: 50.43 mm), and shortest in the BT population (F2,52 = 20.75, r2 = 0.45, P <  0.0001, Fig. 3A). TBW varied among populations, and was smallest in the YN and FN populations, which had similar TBWs (YN: 7.20 mm; BT: 6.93 mm), and the largest in the FY population when controlling for SVL using ANCOVA (F2,52 = 6.82, P = 0.002, Fig. 3B).

Figure 3 Comparisons between (A) snout-vent length and (B) tail width at base of gravid females in three populations (BT, Bei Tun; FY, Fu Yun; YN, Yi Ning) of P. helioscopus.

Points indicate means with 95% confidence intervals. Different letters indicate significant differences at the P < 0.05 level.

Female reproductive traits

Females in the FY population laid heavier eggs than those in the BT and YN populations (Table 1). Eggs were similar in length in all three populations. Eggs were wider in the FY population, and narrower in the YN population. BT females laid smaller clutches than FY and YN females when controlling for SVL (Table 1).

Table 1 Descriptive statistics of female reproductive traits in three populations (BT: Bei Tun, FY: Fu Yun, YN: Yi Ning) of P. helioscopus.

	BT (n = 35)	FY (n = 90)	YN (n = 63)	F-level and P-value	
EM (g)†	0.51 ± 0.02b	0.61 ± 0.02a	0.55 ± 0.01b	F2,187 = 11.67, r2= 0.11, P <0.0001	
range	0.32  ∼ 0.76	0.27 ∼ 1.02	0.28 ∼ 0.82		
EL ( mm)#	15.7 ± 0.24a	14.4 ± 0.17a	14.9 ± 0.16a	F2,187 = 1.15, P = 0.318	
range	12.47 ∼ 18.51	11.49 ∼ 19.50	9.94 ∼ 17.35		
EW (mm)#	8.4 ± 0.08b	8.5 ± 0.06a	8.3 ± 0.07b	F2,187 = 19.42, P <0.0001	
range	7.19  ∼ 9.03	6.90  ∼ 9.90	6.39 ∼ 9.36		
ES†	1.8 ± 0.03a	1.7 ± 0.02b	1.8 ± 0.02ab	F2,187 = 6.71, r2 = 0.06, P <0.0001	
Range	1.44 ∼ 2.27	1.43 ∼ 2.18	1.47 ∼ 2.11		
CS#,*	2.9 ± 0.13b	3.7 ± 0.18a	3.8 ± 0.14a	F2,187 = 10.93, P = 0.0001	
range	2  ∼ 4	2  ∼ 6	3  ∼ 5		
Notes.

EM egg mass

EL egg length

EW egg width

ES egg shape

CS clutch size

Different letters indicate significance at the P < 0.05 level.

† ANOVA.

# One-way analyses of covariance (ANCOVAs) (for CS with snout-vent length (SVL) as the covariate, and for EL and EW with egg mass (EM) as the covariate).

* BT n = 13, FY n = 24, YN n = 16.

Egg size-clutch size trade-offs

We found a positive relationship in all populations between EL and EM (Fig. 4C). In BT and FY populations, egg size decreased with clutch size, while in YN females, egg size was independent of clutch size (Figs. 4A, 4B).

Figure 4 Regressions of egg mass (EM) (A) and egg length (EL) (B) and clutch size (CS) trade-offs, and relationship between EM and EL (C) in P. helioscopus.

Fitted reduced major axis regression model and statistical significance (P < 0.05) are indicated in each case. BT, Bei Tun–Asterisk; FY, Fu Yun–Triangle; YN, Yi Ning–Circle. Points were jittered using the geom_jitter function.

Relationships among egg size, clutch size, and female morphology

In the BT and YN populations, female body size was independent of EL, EW, and CS (Fig. 5). In the YN population, while CS was independent of female body size (Fig. 5C), EL was weakly correlated with SVL (Fig. 5A), and EW and TBW were correlated (Fig. 5B).

Figure 5 Regressions of egg length (EL) (A), egg width (EW) (B), and clutch size (CS) (C) and female morphological traits (SVL, snout-vent length; TBW, tail base width) from three populations of P. helioscopus.

Fitted reduced major axis regression model and statistical significance (P < 0.05) are indicated in each case. BT, Bei Tun–Asterisk; FY, Fu Yun–Triangle; YN, Yi Ning–Circle. Points were jittered using the geom_jitter function.

Discussion

Females in our three populations varied in body size (SVL, TBW), and reproductive traits (EM, CS, RCM, and egg size), in ways that supported, with some exceptions, an egg size-clutch size trade-off in the three populations of P. helioscopus. Female body size, egg size, and clutch size were smaller in the BT population than in the FY and YN populations, and the FY and YN populations laid more, and rounder eggs. Egg size was not correlated with female body size in the BT and FY populations, but egg size-clutch size trade-offs occurred in both populations. Trade-offs between egg size and clutch size were not found in the YN population, but egg size was correlated with female body size in this population.

Morphological traits, such as body size and body shape, always vary among different populations of animals (e.g., snakes: Zhong et al., 2017); lizards: (Horváthová et al., 2013); turtles: (Werner et al., 2016). Environmental factors that exert strong effects on animal life history traits include activity season length and food availability (Yom-Tov et al., 2006; Horváthová et al., 2013). Our study revealed that females in the FY and YN populations were larger than females in the BT population. Longer activity seasons were assumed to be the cause of variation in the body size between the females of the YN and BT populations (Liang & Shi, 2017). Temperature, which is fundamentally important for lizards (Grant & Dunham, 1990), was highest in YN, especially in March and November (Fig. 2A). In YN, P. helioscopus activity began in mid-March, and hibernation began in early November, which means that the activity period for lizards in YN was almost a month longer than in the other two sites (BT and FY). P. helioscopus in this population were larger, probably because of the longer growing season. It is possible that P. helioscopus age varied between the three study locations, and this may have influenced results. However, we have no reason to believe that the ages of the P. helioscopus captured from the three populations were very different. We could have tested for this possibility using mark and recapture methods, but unfortunately, this was outside the scope of this study.

The BT and the FY populations experienced similar temperatures, which raises the question as to what caused the differences in body size. One possibility is that food limitation might have resulted in reduced growth rates in the BT population. Rainfall is critical for habitat quality (e.g., vegetation cover and prey abundance (Lorenzon, Clobert & Massot, 2001) and there was significant geographic variation in rainfall between BT and FY (Fig. 2). BT is sparsely vegetated, due to drier conditions in this location, whereas the vegetation is more abundant in the wetter FY and YN sites (Figs. 1, 2B). Humidity is the most important factor influencing the abundance and distribution of insects (Savopoulou-Soultani et al., 2012; Cesne, Wilson & Soulier-Perkins, 2015), and therefore drier conditions and sparse vegetation could be associated with less available food in the BT population.

Egg size varies among populations because of variation in the female body size, which is an important female trait that can affect offspring quality (Steyermark & Spotila, 2001; Morrison & Hero, 2003; Olsen & Vøllested, 2003). We found that egg size differed among the three populations, which suggests that larger females in the FY and YN populations were able to allocate more resources to egg production. In addition, egg size was also correlated with the incubation period, with smaller eggs having a relatively shorter incubation time. However, further studies are needed to determine whether the earlier hatching of the smaller eggs in the BT population could provide offspring with more time to forage before the hibernation period (Thompson & Pianka, 2001). EL was similar in the three populations (using EM as the covariate), but EW was not. However, egg shape is also related to clutch size, and larger clutches tend to have more rounded eggs (Ji et al., 2002). Eggs were narrower in the BT population, and both the FY and YN populations laid more, and rounder eggs (Table 1). Larger reptile females tend to lay more eggs (Ryan & Lindeman, 2007; Amat, 2008). Thus, the smaller CS of the BT population, associated with their smaller body size, may be due to limited resources (including food), as resource availability often varies among populations (Liao, Lu & Jehle, 2014; Roitberg et al., 2015).

The trade-off between egg size and clutch size is an important concept in life-history theory (Kern et al., 2015). Egg size (EM and EL) and clutch size were negatively correlated in the BT and FY populations, but not in the YN population. In the YN population, there was no egg size-clutch size trade-off, suggesting that patterns of variation in egg size and clutch size are not always consistent in different populations (Liao, Lu & Jehle, 2014; Roitberg et al., 2015).

Generally speaking, offspring phenotypes are influenced by female body size (e.g., SVL, Krist & Remeš, 2004). Body size and other factors affecting egg size will result in the following five possible outcomes (Lovich et al., 2012): (1) egg size is constrained by female size (not optimized), (2) egg size is unconstrained by female morphology (optimized), (3) egg size is unconstrained by female morphology and optimized only in the largest females (Fehrenbach et al., 2016), (4) egg size is not constrained by the pelvic aperture width, and is not optimized, but rather is constrained by some other non-morphological factor (e.g., female age or clutch number Clark, Ewert & Nelson, 2001; Harms et al., 2005; Paitz et al., 2007), or (5) egg width is constrained and requires osteokinesis for oviposition (Hofmeyr, Henen & Loehr, 2005; Fehrenbach et al., 2016).

Consistent with the predictions of the morphological constraint hypothesis, egg size increased as the size of the female increased (outcome 1) in the YN population. Although female body size in the BT population was smaller than that in the FY population, in both cases, egg size was not correlated with female body size (outcome 2 or 4 above). For some species with small body sizes, egg size is constrained by female morphology (Ryan & Lindeman, 2007). In small-bodied females, the body size-specific constraints on egg size, coupled with selection towards an optimum egg size, results in a positive correlation between body size and egg size. Egg size (EL and EW) was not dependent on female body size in either the BT or FY population, but there were negative correlations between egg size and clutch size (Fig. 4), suggesting that egg size was constrained by CS (a non-morphological factor) in both populations (Brown & Shine, 2009; outcome 4). Unexpectedly, we found that egg size was correlated with body size in the larger-bodied females of the YN population. A positive relationship between egg size and female size indicates that there is no optimal egg size in the YN population (Escalona, Adams & Valenzuela, 2018). However, we found some support for the prediction that EW is constrained by TBW (Fig. 3), since eggs must fit the female tail base width, which they pass through on their smallest axis (i.e., EW). In some turtle species, EW but not EL increases with the size of the female (Rasmussen & Litzgus, 2010). There was a significant (but weak) positive correlation between EL and female SVL in the YN population, suggesting that EL is dependent upon on female SVL. EL can be constrained by morphological factors, non-morphological factors (e.g., CS), or their interactions, which may indicate that a weak relationship exists between female morphology and EL in the YN population. The specific mechanisms of the non-morphological factors require further study (Kern et al., 2015).

Conclusions

We found geographical variation in the body size and reproductive strategies of P. helioscopus. Females in populations with longer growing seasons and abundant vegetation (the FY and YN populations) were larger. Lizards in the BT population were smaller, perhaps due to limitations on food availability, or limitations on activity seasons, and also had smaller clutches than the FY and YN populations. Females in the FY and YN populations produced rounder eggs, perhaps due to their larger body size. This study found that egg size was correlated with female body size in the larger-bodied females of the YN population—an anomaly for the morphological constraint hypothesis. Egg size was not correlated with female body size, and did not follow the optimal egg size hypothesis in the BT and FY populations. Trade-offs between egg size and clutch size suggest that egg size was constrained by clutch size in both populations.

Potential genetic and age variation associated with females in these populations may have influenced our results. However, here we demonstrate that life histories, as measured by body size and clutch characteristics, can vary in surprising ways, sometimes supporting the possibility of trade-offs, and sometimes not.

Supplemental Information

Table S1 Trait values of egg size, clutch size, and female morphology

Click here for additional data file.

Table S2 Trait values of clutch size, clutch mass, relative clutch mass, and female morphology

Click here for additional data file.

Supplemental Information 1 R codes and data

Click here for additional data file.

We are grateful to James Roper, Prof. Lovich of the United States Geological Survey, the Southwest Biological Science Center, and anonymous reviewers for their excellent suggestions for improving this manuscript. We thank Luo D and Wang P for assistance during fieldwork, and we thank An J for help with the egg collection and lizard husbandry. Mr. T Martin provided professional advice regarding spelling and phrasing.

Additional Information and Declarations

Competing Interests

Author Contributions

Animal Ethics

Data Availability

The authors declare there are no competing interests.

Tao Liang conceived and designed the experiments, performed the experiments, analyzed the data, contributed reagents/materials/analysis tools, prepared figures and/or tables, authored or reviewed drafts of the paper, approved the final draft.

Lu Zhou conceived and designed the experiments, performed the experiments, analyzed the data, prepared figures and/or tables, authored or reviewed drafts of the paper, approved the final draft.

Wenfeng He performed the experiments, contributed reagents/materials/analysis tools, prepared figures and/or tables, approved the final draft.

Lirong Xiao performed the experiments, contributed reagents/materials/analysis tools, approved the final draft.

Lei Shi conceived and designed the experiments, performed the experiments, analyzed the data, contributed reagents/materials/analysis tools, authored or reviewed drafts of the paper, approved the final draft.

The following information was supplied relating to ethical approvals (i.e., approving body and any reference numbers):

Specimens were collected following Guidelines for Use of Live Amphibians and Reptiles in Field Research (the Herpetological Animal Care and Use Committee (HACC) of the American Society of Ichthyologists and Herpetologists, 2004). This study was conducted in compliance with current laws on animal welfare and research in China and the regulations set by the Xinjiang Agricultural University. After the research was completed, the lizards were released where they were captured.

The following information was supplied regarding data availability:

The raw data are provided in the Supplemental Tables.

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
