# Peer review of "Variations in the reproductive strategies of three populations of Phrynocephalus helioscopus in China"

_PeerJ, doi:10.7717/peerj.5705_

## Round 0.1 · original submission · Major Revisions

Both reviewers found interesting data and ideas, but with reservations. As I read the paper, I have additional observations and comments. I'll comment on the lesser issue first - the English. For the most part, it is understandable, but the text in many places is awkward. I'll give a couple of quick examples - beginning at line 77.

Your first line - 1. Determining whether variations exist in female snout-vent length (SVL) and tail base width (TBW) between the three populations.

You should really mean to say - To test whether reproductive female size differs in the three populations. (Here, you do NOT need to state which variables you will use to test this - clearly you DO need to talk about it, but that can come later). Also, you must explain why female size MIGHT be the same in 3 geographically different populations - there are many reasons to vary, fewer to be the same - and your (assumed) null hypothesis is that they are the same.

79 - The list of variables should be of independent variables and not as if you measured many variables when some are simply consequences of the others. So,

2. If size varies between populations, to test how that variation is associated with reproductive traits, especially:
a. fecundity
b. egg and clutch size and clutch mass (but not, clutch size and fecundity are two ways of saying almost the same thing. Also, clutch mass is simply a consequence of egg and clutch size, so adding it as a new variable seems strange).
c. how female size is associated with the measures of egg and clutch size.

It almost seems like you could have said:

2. How female size varies with respect to fecundity, which will be measured as egg size, clutch size, and clutch mass.
3. Whether those relationships (female size and fecundity) suggest differential selective pressures unique to each region.

Or something along those lines. If you can, I would suggest you find a good English speaker who can help you write more succinctly.

In talking about egg shape, there is no reason to refer to a round egg as "width biased." Indeed, you may simply refer to eggs as round to elongate.

Your first sentence of your results is:
The mean value for female SVL differed significantly among the three populations (F2,52=20.75, P < 0.0001, One-way ANOVA).

It would be more succinct and better written if you wrote:

SVL varied between populations and was longest in the similar YN and FN populations (here STATE the sizes), and shortest in the BT population (Statistics).

You will note I subsumed the first two sentences into one.

Figure one show three study populations, but in the legend you state that they are indicated by black dots while your figure also has three red dots that you do not explain. Also, the legend is incomplete in general. Something like the following would be better:

Figure 1. Map, showing the three locations where lizards were captured for this study in the Xinjiang Uyghur Autonomous Region of western China. Closest cities (Beitun - BC, Fuyun - FC, and Yining - YC) are identified by the red dots, and the collecting locations are indicated by the black dots with arrows. Photos indicate habitat types in each sampling location.

I notice that in your figure 2, you seem to use Tukey to compare locations, using letters to identify differences. However, you do NOT state what the intervals around each point are. Are they confidence intervals? They do not seem to be, but it would be better if they were. Also, in panel A, you use upper-case letters, while in B you use lower-case letters. Please clarify and be consistent. The legend is also poorly worded. Here is a suggestion:

Figure 2. Comparisons between A) snout-vent length and B) tail width at base, of gravid females in three populations of P. helioscopus. Letters indicate Tukey tests where different letters indicate different values. Points are means with 95% confidence intervals.

Also, you can start the Y axis in B at 6.5 instead of 6. And, because you use log in other places, should you not use a log10 axis in these figures? Finally, the confidence intervals do not agree with the letters - something is amiss, so you should clarify. For example, the two A in A do not overlap at all, and they are the same. In B, on the left two values (b, a), they DO overlap, yet they are different. These are inconsistent and for this reason, I think you should use 95% confidence intervals. Also, you do not need lines on the right and top of each figure.

Next, in Figure 3, your panels A and B are the exact same, while the variables are not. Also, in both, the lines are not explained, and to me, they do NOT seem to fit the data. In fact, because you have 3 populations, you should also have three regression lines. You should also explain the lines in the figures or in the legend. Finally, there is no reason to use log of Clutch size, because, clearly, it does not vary logarithmically.

Again, the legend is incomplete in Figure 4. You are saying correlation, but your lines suggest you are using regression - they are not the same. Also, why is it that there is no line for the YN population? What is the confidence interval of the relationship? What is the relationship between egg weight and egg length? These relationships do not appear to be linear and I don't think your residuals are normally distributed. Can you clarify these?

In your table, you provide fractions of seconds for locations, but you really neither have nor need such precision. I'd drop the fractions. And, while these values LOOK different (rainfall, temperature and so on, you provide no statistical tests and all of these values could be statistically similar. Can you show a graph of these values over the course of the year to better illustrate how climate varies?

I recommend that you clarify your statistics throughout the paper. Both reviewers and I agree that there are issues. Are the assumptions correct? Are the statistics valid? Are the relationships linear? Exactly what is the nature of the tradeoffs?

Reviewer 1 ·

Basic reporting

Below are a number of specific places I see that the authors could focus on to help improve the presentation of their work. (A few of these may be more of a language issue.)
-Climate data is not developed in the methods in any detail, nor in the results. It is discussed in generalities with causal implications given in the discussion. This needs to be changed. Temperature is a very important factor in ectotherm life history and so should get some detailed attention.
- More natural history of the species would provide greater context for the results.
line 10: What is “fitness of a reproductive event?”
line 24: there is no real evidence provided for selective pressures. These data provide a description of geographic variation in reproductive traits and morphology but selection pressures implies a difference in fitness which is beyond the current study.
Line 25: correlated yes, “constrained by” I am not convinced.
Line 35-36: who has given them more attention? Need reference here
Line 36-38: Does this have to occur?
Line 41-42: always?
Line 77-87: format issues
Line 84: I do not understand point e. This needs to be clarified.
Line 173: always? This is a snake study reference. Here and throughout need to cite relevant lizard studies and indicate when a general pattern is being presented or something specific to this study system. There are many sweeping statements made in the discussion that may not be true or are specific to a given study system (eg line188-191). In a number of places studies of other organisms are cited as support for the current results but the connection is not clear.
Line 205-207: If size was adjusted for, then it is unclear what the role of body size is here.
Line 239: “some species” refers to?
Figure and Table legends need much more detail. For example, it is not clear what is being shown on Fig 3and 4, what are the lines? In the climate table no clear idea about time frame is given.
-Remove “s” from end of variation throughout.

Experimental design

While there may be some issues with the descriptive statistical analyses they may be fairly sound if assumptions of the models are met and more information is provided/clarified (see specific comments below). As they state early on, there is not much reproductive data for this species. This manuscript begins to take care of that, which is a good and necessary first step for understanding their life histories.
Line 91 - : These populations were sampled during different years. What is the year-to-year variation these reproductive traits within any one site? Did the weather vary over those years/sites?
Line 101- Were the cages checked 12 times per day?
Line 103- how were the measurements taken? Equipment?
Line 106: What is POM?
Line 112: consider cutting the 1st sentence
Line 113-115: This is not clear. Were assumptions met for ANOVA, ANCOVA? These types of traits can show high collinearity which can be a problem for ANCOVA. Were Tukey’s done for ANCOVA and ANOVA?
Line 118: OLS regression – define
Line 119-121: What data were used, for what purpose, and how was it analyzed?
Line 121 how were means adjusted?
No weather/climate data provided in text of results.
Results general: don’t need to show p value cut off with each result.

Validity of the findings

Where the most work appears to be required is in the level of interpretation of the results. The discussion of the results goes well beyond what can be said with the data in hand. Factors causing “life history trait” variation are many and often display complex interactions. The author’s should consider looking at Grant and Dunham (1990) for some ideas about complex interactions in causal agents to life history trait variation across populations of lizard species. Many lizard species also show year to year and time of season variation in these types of reproductive traits (eg clutch size). Therefore, to try to associate the patterns in the reproductive traits to any given causal agent or trade-off with only the descriptions of 1 set of samples from each site taken in different years may be premature.
Discussion:
Line 180-186: All this needs to be developed in a rigorous manner in the methods and results.
These populations appear to be in vastly different habitats and there may be many, many different factors causing any of the patterns presented here. The authors need to step back and be very specific about what they can and cannot say with the data in hand. For example, they discuss higher food availability in one of the locations as a cause of large SVL. There needs to be data to support this, both the level of food availability and causal relationship, as SVL can be impacted by other factors like temperature and activity time.
Line 195: How does hydration state impact egg size/mass? Eggs absorb water in moist soil and can expand after being laid. Could this confound the implications of egg size variation as measured and described?
Line 228- Correlation is not causality. Need to be cautious in interpretation of patterns.
Conclusions: There are many other reasons for variation in these traits. For example: Season length, daily temperature, food availability, daily activity times, social environment, predator environment… Again, caution needs to be used when interpreting the patterns being reported.

Additional comments

This is a potentially interesting description of geographic variation in reproductive traits and some female morphological traits in a wide spread species of lizard.
The authors make a good effort to collect some key reproductive traits across three geographically widely separated populations, each of which appears to be in very different habitats. This could become the basis for an interesting comparative study that builds on these initial findings.

Reviewer 2 ·

Basic reporting

The MS is well-written, and the datasets are large enough to address the questions of interest. The literature review is adequate.

Many of the empirical patterns are well-known from other taxa (e.g., small-bodied females produce elongate eggs simply because there isn't room in a slender oviduct for a rounded egg). However, there is value is having such data for a "new" lineage of lizards, especially because so much of the existing literature is based on taxa from the USA and Europe.

Experimental design

My main suggestion is that at the moment, the comparisons among the three populations are based on whether tests on data for each population (separately) meet or do not meet the standard threshold for "significance" (P < 0.05). It is easy for one dataset to fall below that threshold and another above it even if there is no "real" (statistically significant) difference between them. So to conclude for example that one species shows a maternal-size constraint on egg size whereas another does not, ideally we need to see a significant difference between those taxa in that relationship not just in size-corrected mean values for egg size etc. There are simple ways to evaluate such differences (including absolute values of residual scores from linear regressions, as measures of dispersion).

Validity of the findings

I think the data are useful. The tradeoff between fecundity and egg size is clear. I am less convinced about the exact nature of tradeoff disparities among populations. For example, I find it difficult to imagine maternal body size constraining egg size in a lizard, given the flexibility of the egg and the cloaca, and the ability of selection to fashion elongate eggs. However, I don't see that ambiguity as a crucial issue.

Additional comments

This well-written manuscript analyses a substantial dataset on poorly-known lizards to explore issues of reproductive allometry. Substantial geographic differences in both maternal and egg traits suggest differences in selective forces, but inevitably such hypotheses are difficult to test in any rigorous ways. The paper does a good job of describing the patterns in the data, and suggesting possible interpretations.

---

## Round 0.2 · Major Revisions

I appreciate that you improved several details in the current manuscript, however, with changes, the English has deteriorated a bit and there are other issues as well. Please read carefully the manuscript that I have marked. I chose to not send back to the reviewers because there are too many details to be resolved. That is, I do not wish to cause the reviewers unnecessary effort. Here I will detail some of my main concerns.

1. The English has become complicated and because of that, you have a lot of redundancies in the text. The way you divided the results and discussion, you often repeat talking about variables and relationships because some of your variables are not independent of others. You should just bring all your egg measurements together (as an example) into one part. You have egg length and width, and shape, but shape is just the ratio of length to width, so they are all intertwined. If you talk about the ratio, then you need NOT talk about either length or width, and you can say that when you describe using the ratio. And, you can say that rounder eggs tend to be larger eggs, becuase egg length is apparently constrained. So, the only way an egg can get larger is by becoming rounder. A sphere maximizes volume at any given radius.

2. Your statistics still need some clarification. For example, you provide two levels of P in your anovas, but you do not need to. An alpha of 0.05 is sufficient and will not change your conclusions. So, simplify and make it more readable. Also, you provide regression lines with out equations of the lines (that you could put in the legends of the figures with regressions). BUT, you include lines when the regressions were NOT significant, and you shouldn't. Also, there is no interpretation of r-squared values when P > 0.05 and so it also does not need to be included. Note my comments in the statistics section of the commented manuscript.

3. Your figures are not as enlightening as they could be. Figure 2 has no need for colors and filled spaces when three lines would be sufficient. Also, are these values DURING the study or the long term trends? I would think long term trends are more meaningful because they would show the trends over time to which the lizards are exposed and which should be closer to evolutionary trends, rather than the short term trends of a single year. Fig. 4 includes lines that are probably not statistically significant and so can confuse the reader. For example, in A, it looks like you are saying that as clutch size increases in YN, so does egg weight. But, that would imply that more eggs are larger eggs and is completely contrary to the other two sites. I believe that relationship was not significant - one way to show that would be to include confidence intervals of the regressions. Also, not that the axis of clutch size has fractions, but eggs are never in fractions. You should make the numbers along the axis be in whole numbers, from 2 to 6, not 1.5 to 6.5. And, if possible, you should "jitter" the points because it i s hard to see with all those points overlaid. In Fig. 5, you have in A, log10 SVL, but in C, you have SVL (not log10). If you need to transform in one place, you should in all. I know I suggested the log10 transform, but that was contingent upon the need. You mentioned testing normality of the "data" but did not speak of the residuals, when the residuals require normality. Just looking at the relationships it's hard to see whether they require transformation. Also, in your statistical tests you talk about Kruskal-Wallis and wmc, but the link was bad and it's not clear why you spoke of that and whether it was necessary. This lead to my confusion.

4. In this version, you embedded the tables and figures in the text, unnecessarily, which caused some confusion as I commented on your text. You can put all your figures in sequence in ONE document (one figure per page) and you can simply place tables at the end of the text (after the bibliography, and one table per page). Both would make it much easier to read!

Thus, due to the confusion with the English, the analyses and the figures, I think a major revision is still necessary. I am attaching your manuscript converted to PDF, with my insertions (I used mark changes, so they will be visible) and comments.

---

## Round 0.3 · Minor Revisions

While the English has been considerably improved, the reviewer and I found some additional wording, and logic, that should be worked on a bit further. Please note in my edited copy of your manuscript that I made many suggestions and comments to help with this process.

Also, I am still concerned with your statistics - please see my comments in Figure 4. I believe I mentioned before that you do not need to (and should not) log10 transform CS. I explain in my comments there. Also, figure 4 requires additional adjustment because the text (regression equations) within the figures are too small to be legible once the figures are reduced for publishing. Also, your letters, A and B are nearly hidden due to the lines. ggplot allows you to put them elsewhere or make them larger so to be more obvious. Finally, CS in all figures is problematic, because it should NOT be in fractions. That is, there should be no variation around each value (did you use jitter?). If you used jitter, you should experiment to make each clutch size less variable (and state that in the legend!).

Reviewer 2 ·

Basic reporting

OK, although some of the new text needs correction for English expression.

Experimental design

OK

Validity of the findings

OK. The primary value is the patterns, rather than the interpretation.

Additional comments

The paper is considerably improved.

---

## Round 0.4 · accepted · Accept

Your changes have improved the manuscript quite a bit. There are still a few minor changes I would like you to make in your figures. We have talked about the use of "jitter" in producing the figures that include clutch size. I think you should either eliminate the jitter, or reduce it very much. In your figures, the jitter mistakenly gives the impression that clutch sizes can include fractions and the clearly cannot. This also means you should double check your use of regressions and the normality of the residuals when using clutch size. Your lines in those figures extend well past the distributions of the data points and probably should not. So, in your figures A and B of figure 4, and C of figure 5, you should work on the jitter. In ggplot there is a width function that allows you to make it narrower. Also, it is possible to jitter each species a little differently so that one can see which species is which.

#